# Sex-Based Evaluation of Lipid Profile in Postoperative Adjuvant Mitotane Treatment for Adrenocortical Carcinoma

**DOI:** 10.3390/biomedicines10081873

**Published:** 2022-08-03

**Authors:** Sarah Allegra, Soraya Puglisi, Chiara Borin, Francesco Chiara, Vittoria Basile, Anna Calabrese, Giuseppe Reimondo, Silvia De Francia

**Affiliations:** 1Laboratory of Clinical Pharmacology “Franco Ghezzo”, Department of Clinical and Biological Sciences, University of Turin, San Luigi Gonzaga Hospital, Orbassano, 10043 Turin, Italy; 336124@edu.unito.it (F.C.); silvia.defrancia@unito.it (S.D.F.); 2Internal Medicine, Department of Clinical and Biological Sciences, University of Turin, San Luigi Gonzaga Hospital, Orbassano, 10043 Turin, Italy; soraya.puglisi@unito.it (S.P.); chiara.borin@edu.unito.it (C.B.); basile_vittoria@libero.it (V.B.); anna.calabrese678@gmail.com (A.C.); giuseppe.reimondo@unito.it (G.R.)

**Keywords:** o,p’-DDD, o,p’-DDE, menopause, dyslipidemia, side effect, safety, sex, gender, cholesterol, triglycerides

## Abstract

Background: A wide interindividual variability in mitotane concentrations and treatment-related dyslipidemia have been reported. Here, we aimed to underline the sex-related differences in the lipid profile in patients that underwent radical surgery of adrenocortical carcinoma during treatment with adjuvant mitotane. Methods: A chromatographic method was used to quantify the drug in plasma collected from adult patients with complete tumor resection, also considering active metabolite o,p’-DDE. Results: We observed different lipid profiles between males and females and between pre- and post-menopausal women. Considering the mitotane-related effects on lipid levels, we observed that higher drug concentrations were correlated with higher HDL in all the considered groups (*p* < 0.001), with total cholesterol both in males (*p* = 0.005) and females (*p* = 0.036), with triglycerides in postmenopausal females (*p* = 0.002) and with LDL in male patients (*p* < 0.001). Increases in o,p’-DDE were positively correlated with HDL levels in all the groups (*p* < 0.001) and negatively with LDL in all the groups (males *p* = 0.008, pre- and post-menopausal females *p* < 0.001), with total cholesterol in pre- (*p* = 0.016) and post-menopausal women (*p* = 0.01) and with triglycerides in premenopausal females (*p* = 0.005). Conclusions: This is the first study designed to evaluate sex differences in lipoprotein and lipid levels during mitotane adjuvant treatment; the results suggest that a gender and personalized approach could be useful to prevent and manage alterations in the lipid profile.

## 1. Introduction

Mitotane (ortho,para,dichlorodiphenyl dichloroethane, o,p’-DDD, 2,2-bis(2-chlorophenyl-4-chlorophenyll-l,l-dichloroethane) is an adrenolytic drug derived from the insecticide dichlorodiphenyltrichloroethane, known since 1959 for its effects on the adrenal cortex [1]. In the mitochondria of adrenal cells, mitotane is metabolized in o,p’-DDE (1,1-(o,p’-Dichlorodiphenyl)-2,2 dichloroethene) through α-hydroxylation and o,p’-DDA (1,1-(o,p’-Dichlorodiphenyl) acetic acid) through β-hydroxylation [2]. Although it is known that mitotane causes the apoptosis of adrenal cells through the deregulation of cytochromes P450 enzymes, the depolarization of mitochondrial membranes and the accumulation of free cholesterol, its exact mechanism of action is still inadequately elucidated at the molecular level due to conflicting findings from in vitro studies [3].

For its apoptotic effect specifically directed to adrenal cells, mitotane is the reference drug for metastatic adrenocortical carcinoma (ACC), alone or in combination with chemotherapy, and it is recommended for adjuvant therapy in patients affected by localized ACC with high risk of recurrence after surgery [4,5]. In both these settings, mitotane therapeutic drug monitoring is recommended to optimize the benefit and to limit adverse events [6]. In fact, many studies have demonstrated survival advantages in patients whose plasma mitotane concentrations were >14 mg/L [7,8,9,10,11,12]. On the other hand, plasma mitotane levels >20 mg/L seem to be associated with central neurologic toxicity (cerebellar symptoms, impaired cognitive performance). However, also in patients with plasma mitotane concentration in the target range (14–20 mg/L), many side effects of mitotane have been reported, including the expected hypoadrenalism in all patients and, in more than one third, central hypothyroidism, hypogonadism in men and the development of ovarian cysts in women [13]. Mitotane has also been found to increase the levels of cortisol binding globulin (CBG) and sex hormone-binding globulin (SHBG) [13,14,15,16]. Moreover, mitotane has a profound impact on lipid levels, causing a marked increase in total, LDL and HDL cholesterol levels in more than 50% of patients, reversible after mitotane discontinuation [13,14,17].

This effect is mediated by the stimulation of 3-hydroxy-3-methylglutarate-coenzyme A reductase (HMG-COA reductase), leading to increased cholesterol synthesis [18]. At the same time, the lipoprotein profile may influence mitotane drug distribution [19]. In dyslipidemic patients, high plasma mitotane levels have been observed, without reported side effects, highlighting that plasma mitotane distribution in lipoprotein could be the major determinant of its distribution in the other tissues [20]. In addition, mitotane bioavailability depends on different factors, such as gender [21]. Sex differences in drug pharmacodynamics and pharmacokinetics, response to treatment and related toxicity have been reported [22].

The objective of the present project was the evaluation of the interplay between mitotane pharmacokinetics, and lipoprotein and lipid levels in patients with complete tumor resection. Mitotane and its metabolite o,p’-DDE were quantified in human plasma, and the biologic implication of serum lipoprotein on drug pharmacokinetics and efficacy were explored considering sex-related differences and the menopausal period.

## 2. Materials and Methods

### 2.1. Patients and Inclusion Criteria

We enrolled ACC patients treated at San Luigi Gonzaga University Hospital. All of the participants underwent radical surgery for ACC and started mitotane adjuvant treatment. The considered inclusion criteria were: age ≥ 18 years, histological diagnosis of ACC (Weiss score [23]), ENSAT stage I–III, complete tumor resection (R0 = free resected margins; R1 = microscopic involvement of resected margins; RX = not determined), postoperative follow-up information and regular mitotane therapeutic drug monitoring. Exclusion criteria were: incomplete resection (R2 = macroscopic invasion of resected margins), ENSAT stage IV, concomitant diagnosis of cancers in the previous 5 years (except for non-melanoma skin cancer treated radically), concomitant disease (also including diabetes, hypertension), less than 6 months of follow-up duration, start of mitotane treatment more than 6 months after surgery, concomitant chemotherapy or radiotherapy and statin or other lipid-lowering treatments. In all patients, as per clinical practice in our center [13,14,24], the following tests for the evaluation of endocrine function were periodically carried out during mitotane treatment (usually every 3 months): thyroid stimulating hormone (TSH), thyroxine (fT4), plasma renin activity (PRA), aldosterone, serum cortisol, adrenocorticotropic hormone (ACTH), testosterone, SHBG, levels of luteinizing hormone (LH) and follicle-stimulating hormone (FSH). Patients with hormonal deficiencies due to mitotane-related effects (for example, hypothyroidism or male hypogonadism) were adequately treated with replacement therapies [13,14,24] In this study, the menopausal period was defined only when FSH and LH range values were 16.7–134.8 mIU/mL and 15–62 mIU/mL, respectively. Drug formulation was Lysodren^®^ 500 mg tablets (Laboratoire HRA Pharma, Paris, France). Mitotane dose protocol was 1 g daily and was increased every 4–7 days—up to 8–10 g daily or the maximum tolerated dose [25]. Recorded data were: sex, date of diagnosis, age and tumor hormone secretion at diagnosis, R status after surgery, date of recurrence and last follow-up or death. Date of diagnosis was defined as the date of surgery. Recurrence-free survival (RFS) was considered from the time of initial surgery to the first radiological evidence of recurrence (in months) and the overall survival (OS) from the date of initial surgery to the date of death (in months). We also reported levels of total cholesterol, high-density lipoprotein (HDL), low-density lipoprotein (LDL) and triglycerides, plasma mitotane and o,p’-DDE concentrations, which were analyzed during the adjuvant treatment with mitotane (in case of recurrence, patient monitoring in this study protocol was interrupted). The study protocol (“Pharmacogenetic determinants mitotane pharmacokinetics”; registration code: 19846/2015) was approved by the local ethics committee. Written informed consent for the study was obtained from each enrolled subject.

### 2.2. HPLC Analysis

Plasma mitotane and o,p’-DDE metabolite quantification were obtained from blood samples at the end of the dosing interval and immediately before the next drug-dose intake.

Blood samples were collected in lithium heparin tubes and centrifuged at 1500 rpm for 10 min (4 °C). The quantification was performed using a validated High-Performance Liquid Chromatography–UV method (HPLC-UV) [26]. The used chromatographic column for analyte separation was an RP-C18 column. The internal standard was used for the internal control of the performed analyses.

### 2.3. Statistical Analysis

Considering the descriptive statistics, continuous and non-normal variables were taken as median values. Interquartile ranges (IQRs; quartile 1 and quartile 3) measured data statistical dispersion; frequency and percentage described categorical variables. The Shapiro–Wilk test was used to test normality. The Kolmogorov–Smirnov test was used to evaluate the correspondence of each parameter with a normal or non-normal distribution. Median values described non-normal variables. The Mann–Whitney test was used to compare plasma concentration, sex and menopause presence/absence (level of statistical significance: *p*-value < 0.05); the test compares the mean ranks, and the obtained *p*-value shows the chance that a randomly selected value from the population with the larger mean rank is greater than a randomly selected value from the other population. The Pearson linear correlation coefficient (r) was performed to investigate the strength of the association between mitotane and o,p’-DDE concentrations and the levels of total cholesterol, HDL, LDL and triglycerides (interval of confidence at 95% (IC95%)). The following subgroups were considered: males, females and women before and after the menopause period began. Univariate and multivariate linear regression analyses tested the predictive power of the considered variables on plasma mitotane and o,p’-DDE levels (β-coefficient; IC, interval of confidence at 95%); if the obtained *p*-value was lower than 0.2 in the univariate analysis, the variables were considered for the multivariate analysis (statistical significance at *p*-value lower than 0.05). Statistical tests were performed using IBM SPSS Statistics 22.0 for Windows (Chicago, IL, USA).

## 3. Results

### 3.1. Study Population

We retrieved 551 mitotane measurements from 112 ACC patients with complete tumor resection, 215 from males and 336 from females. The baseline characteristics of our cohort are listed in Table 1 and Table 2.

### 3.2. Effects of Sex on Total Cholesterol, HDL, LDL and Triglycerides

A significant influence of sex on total cholesterol (*p* < 0.001; Figure 1), HDL (*p* < 0.001; Figure 2) and triglycerides (*p* < 0.001; Figure 3) resulted. Total-cholesterol and HDL levels were lower in males than in females, and triglyceride levels were lower in females than in males (Table 1). No statistical significances were observed when evaluating LDL levels.

### 3.3. Effects of Menopause on Total Cholesterol, HDL, LDL and Triglycerides

A significant influence of menopause on total cholesterol (*p* = 0.022; Figure 4), HDL (*p* < 0.001; Figure 5), LDL (*p* = 0.002; Figure 6) and triglycerides (*p* < 0.001; Figure 7) was observed. The HDL levels were lower in postmenopausal than in premenopausal women, and total-cholesterol, LDL and triglyceride levels were lower in premenopausal than in postmenopausal women (Table 1).

### 3.4. Effect of Menopause on Total Cholesterol, HDL, LDL and Triglycerides

A significant influence of menopause on total cholesterol (*p* = 0.022; Figure 4), HDL (*p* < 0.001; Figure 5), LDL (*p* = 0.002; Figure 6) and triglycerides (*p* < 0.001; Figure 7) was observed. HDL levels were lower in postmenopausal than in premenopausal women, and total-cholesterol, LDL and triglyceride levels were lower in premenopausal than in postmenopausal women (Table 1).

### 3.5. Correlations between Mitotane and o,p’-DDE Levels, and Total Cholesterol, HDL, LDL and Triglycerides in Male Patients

The Pearson correlation test showed statistically significant correlations between mitotane levels, and total cholesterol, HDL and LDL (Table 2). Considering o,p’-DDE, significant correlations were observed with HDL and LDL (Table 2).

### 3.6. Correlations between Mitotane and o,p’-DDE Levels, and Total Cholesterol, HDL, LDL and Triglycerides in Female Patients

The Pearson correlation test showed statistically significant correlations between mitotane levels, and total cholesterol, HDL and triglycerides (Table 2). Considering o,p’-DDE, significant correlations were observed with total cholesterol, HDL, LDL and triglycerides (Table 2).

### 3.7. Correlations between Mitotane and o,p’-DDE Levels, and Total Cholesterol, HDL, LDL and Triglycerides in Premenopausal Female Patients

The Pearson correlation test showed a statistically significant correlation between mitotane levels and HDL (Table 2). Considering o,p’-DDE, significant correlations were observed with total cholesterol, HDL, LDL and triglycerides (Table 2).

### 3.8. Correlations between Mitotane and o,p’-DDE Levels, and Total Cholesterol, HDL, LDL and Triglycerides in Postmenopausal Female Patients

The Pearson correlation test showed statistically significant correlations between mitotane levels, and HDL and triglycerides (Table 3). Considering o,p’-DDE, significant correlations were observed with total cholesterol, HDL and LDL (Table 3).

## 4. Discussion

Mitotane is the only adrenolytic drug approved for ACC adjuvant treatment. It exerts its adrenolytic action through the alterations in the function of mitochondria and by blocking adrenal steroid 11-β-hydroxylation [3]. In this study, we investigated the possible influence of mitotane plasma concentration on dyslipidemia in patients with complete tumor resection, separately evaluating man, females, and premenopausal and postmenopausal women. We included in this study only patients treated with adjuvant mitotane therapy, because the fact that they were free of disease during monitoring avoided the influence of a possible tumor hormonal secretion. We observed higher total-cholesterol and HDL levels in females, with differences considering the menopausal period, i.e., low cholesterol and high HDL during the postmenopausal period. In premenopausal women, lower triglycerides and LDL were seen. Males reported higher triglycerides. Considering mitotane-induced dyslipidemia, we observed that higher drug concentrations were correlated with higher HDL levels in all the considered groups, suggesting a treatment effect on this lipoprotein independent from sex and female hormonal status. Moreover, higher mitotane concentrations were positively correlated with raised total cholesterol in males and females, triglycerides in females and postmenopausal group and LDL only in male patients. It is already known that the mitotane-induced lipid pattern is characterized by the increase in HDL, rather than LDL cholesterol, and a concomitant rise in triglycerides [13,14,17,18]. In 1981, Stacpoole and colleagues investigated the mitotane endogenous cholesterol production mechanism in rats and in isolated rat hepatocytes, examining the drug effect on HMG-COA reductase. They observed that mitotane stimulated HMG-CoA reductase activity in both in vivo and in vitro models [27]. In 2015, Hescot et al. evaluated the in vivo and in vitro biological implications of serum lipoproteins on mitotane’s pharmacological activity. They quantified the mitotane levels in plasma and in adrenal-tissue samples and analyzed the effect of lipoprotein-bound or lipoprotein-free mitotane on the proliferation and apoptosis of human adrenocortical H295R cells. Moreover, they evaluated statin’s effect on mitotane treatment. They observed that mitotane was widely distributed in each lipoprotein subfraction and that there was a positive correlation between mitotane levels in plasma and those bound to LDL and HDL. Intratumor mitotane concentrations, in 5 of 23 ACC samples of mitotane-treated patients, resulted independent of cholesterol transporter expression, scavenger receptors and LDL receptors. In vitro analyses showed higher antiproliferative and pro-apoptotic effects and mitochondrial uptake of mitotane when H295R cells were grown in lipoprotein-free media. Eventually, statin treatment, together with mitotane administration, was significantly related to a higher rate of tumor control [28]. The HDL-increase mechanism is less clear. Probably, mitotane induces the liver cytochrome P450 activity, enhancing HDL synthesis [29]. Moreover, a possible role of the estrogen-like activity of the drug may be involved [30,31,32]. In addition, mitotane determines the inhibition of scavenger receptor class B member 1 (SR-B1), localized in liver and in steroidogenic organs, involved in HDL selective uptake [33]. Total-cholesterol increase could be related to the drug inhibition of CYP11A1, which leads to a reduction in the conversion rate of cholesterol to pregnenolone [34]. In addition, raised cholesterol levels could be the consequence of hypothyroidism, another mitotane-induced endocrine toxicity [35].

When evaluating the active-metabolite levels, o,p’-DDE increase was positively correlated with HDL levels in all the groups, confirming the results on mitotane. Conversely, negative correlations with LDL in all the groups, with total cholesterol in females and in pre-/post-menopausal women and with triglycerides in females and premenopausal patients were noted. Although both HDL and LDL were associated with o,p’-DDE, different studies reported a stronger association with LDL and very-low-density lipoprotein (VLDL) than with lipoproteins with a larger diameter (HDL) [36,37,38]. This disparity could be explained by the differences in the metabolic pathways of the lipoprotein groups. While VLDL, IDL and LDL are transported from the liver, through the endogenous pathway, to deposit lipids into other tissues, HDL transport lipids from tissues, through the reverse transport pathway, to the liver [39]. The reduced LDL observed when the metabolite concentration increased could probably be due to the increase in LDL-bound o,p’-DDE, not quantified in this study. The used chromatographic method only quantified free plasmatic o,p’-DDE and not that bound to lipoprotein [26]. The same reason could justify the negative correlation between the metabolite and triglycerides; a significant association between serum triglycerides and DDE was observed in adults from the Native American community of Akwesasne Mohawks, from China and from Europe [40,41,42,43]. In addition, a previous study suggested that total cholesterol in lipoprotein is mostly composed of cholesterol esters, which drive the association between total cholesterol and o,p’-DDE [44]. This could explain the negative correlation observed in all the female groups in our study; high levels of LDL-bound o,p’-DDE imply a major use of cholesterol to make up for lipoprotein, resulting in lower levels. Probably, mitotane treatment acts by altering lipid transport mechanisms or the function of biochemical markers involved in lipid synthesis and degradation.

It is already known that males and females respond differently to dietary fat and cholesterol and that menopause affects weight [45,46]. Probably, the body mass index affects subjects differently due to their hormonal differences. Menopause is characterized by lean-muscle mass reduction and total-fat mass increase, predisposing women to obesity [47,48]. Moreover, a worsening of the lipid profile in postmenopausal women was reported, including total-cholesterol, LDL and triglyceride increases and HDL reduction [49,50,51,52]. Thus, menopause lipid and lipoprotein profiles more closely resemble those reported in men. The lipid profile, including total-cholesterol, HDL, LDL and triglyceride levels, should be performed every 3–4 months [4].

## 5. Conclusions

This is the first study designed to separately evaluate in males and in females and in premenopausal and postmenopausal women cholesterol, HDL, LDL and triglyceride levels in ACC patients adjuvantly treated with mitotane after radical tumor resection. The obtained results suggest that a gender and personalized approach could be useful to prevent and to better control dyslipidemias. Further studies, including data about blood lipid levels from healthy volunteers, the comparison between lipid levels before and after the start of therapy with mitotane, drug dose, male hormonal phase, body weight and body-fat distribution are indispensable to clarify the mechanism of mitotane-induced dyslipidemia and in particular hypercholesterolemia.

## Figures and Tables

**Figure 1 biomedicines-10-01873-f001:**
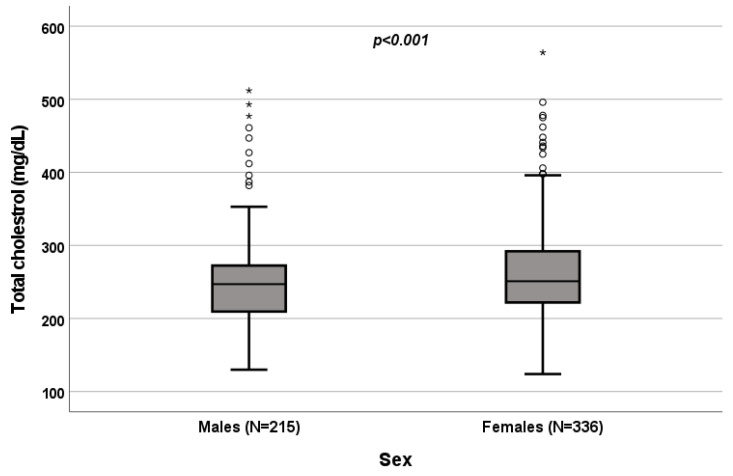
Influence of sex on total-cholesterol levels (mg/dL). Box plot of sex’s influence on total-cholesterol levels (mg/dL): The black lines in the boxes show the interquartile ranges (IQRs): quartile 1, median values and quartile 3; the open dots and stars describe the outlier values. The *p*-value is shown.

**Figure 2 biomedicines-10-01873-f002:**
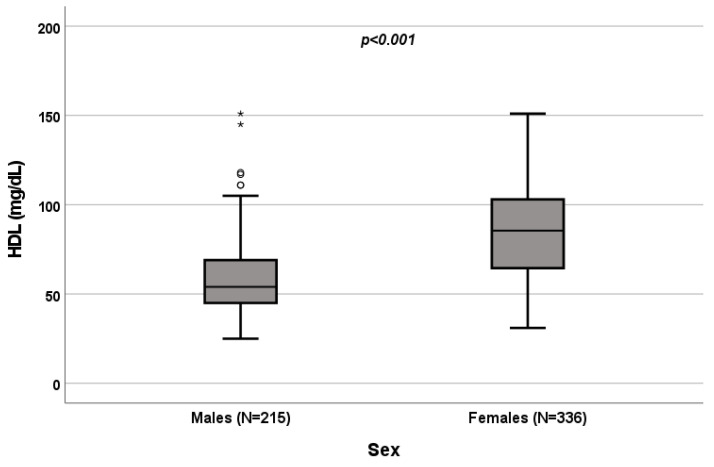
Influence of sex on HDL levels (mg/dL). Box plot of sex’s influence on HDL levels (mg/dL): The black lines in the boxes show the interquartile ranges (IQRs): quartile 1, median values and quartile 3; the open dots and stars describe the outlier values. Symbols (star and circle) indicate outliers of the study sample. The *p*-value is shown.

**Figure 3 biomedicines-10-01873-f003:**
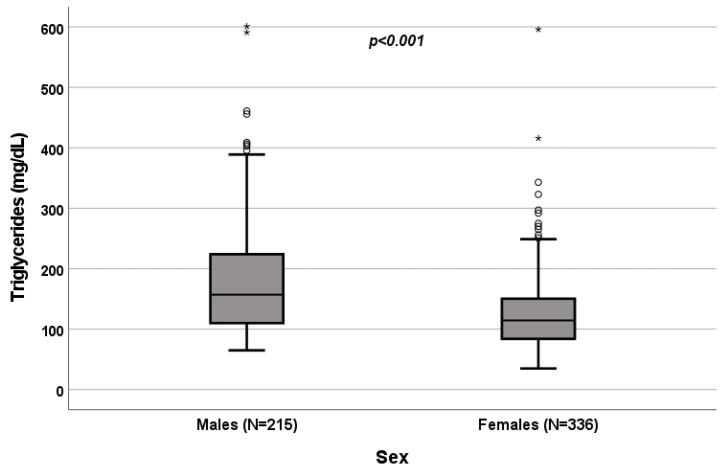
Influence of sex on triglyceride levels (mg/dL). Box plot of sex’s influence on triglyceride levels (mg/dL): The black lines in the boxes show the interquartile ranges (IQRs): quartile 1, median values and quartile 3; the open dots and stars describe the outlier values. The *p*-value is shown.

**Figure 4 biomedicines-10-01873-f004:**
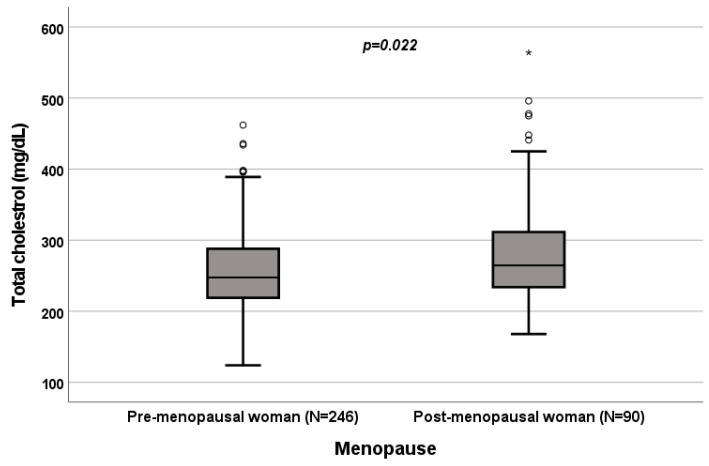
Influence of menopause on total-cholesterol levels (mg/dL). Box plot of sex’s influence on total-cholesterol levels (mg/dL): The black lines in the boxes show the interquartile ranges (IQRs): quartile 1, median values and quartile 3; the open dots and stars describe the outlier values. The *p*-value is shown.

**Figure 5 biomedicines-10-01873-f005:**
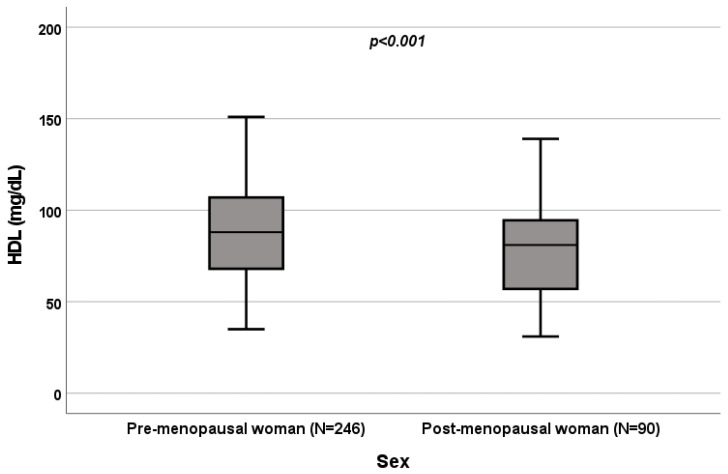
Influence of sex on HDL levels (mg/dL). Box plot of sex’s influence on HDL levels (mg/dL): The black lines in the boxes show the interquartile ranges (IQRs): quartile 1, median values and quartile 3. The *p*-value is shown.

**Figure 6 biomedicines-10-01873-f006:**
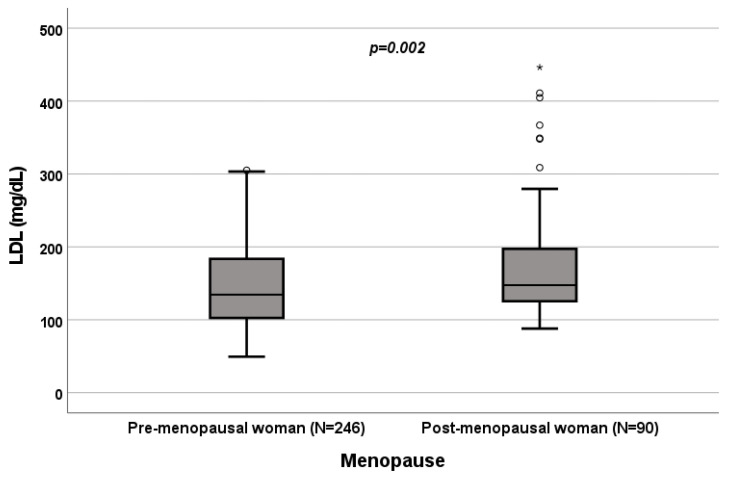
Influence of sex on LDL levels (mg/dL). Box plot of sex’s influence on LDL levels (mg/dL): The black lines in the boxes show the interquartile ranges (IQRs): quartile 1, median values and quartile 3; the open dots and stars describe the outlier values. The *p*-value is shown.

**Figure 7 biomedicines-10-01873-f007:**
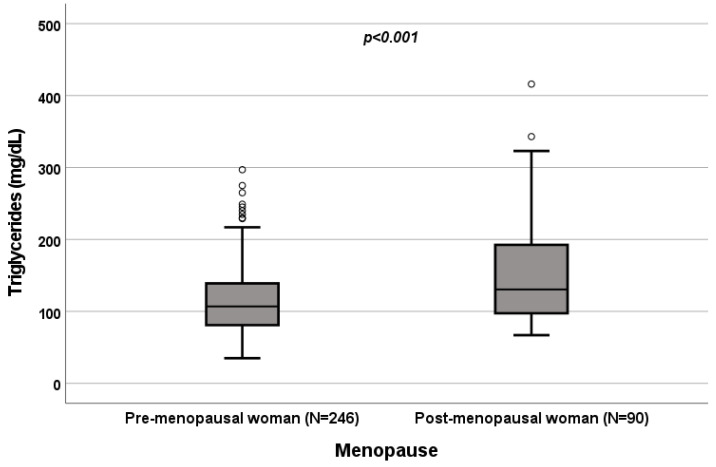
Influence of sex on triglyceride levels (mg/dL). Box plot of sex’s influence on triglyceride levels (mg/dL): The black lines in the boxes showed the interquartile ranges (IQRs): quartile 1, median values and quartile 3; the open dots describe the outlier values. The *p*-value is shown.

**Table 1 biomedicines-10-01873-t001:** Demographic and clinical characteristics of the 112 enrolled patients.

Variable	All (*n* = 112)	Males	Females (*n* = 74)	Premenopausal Women (*n* = 50)	Postmenopausal Women (*n* = 23)
		(*n* = 38)			
**Age (years)**					
Median (IQR)	43.21	40.97	44.37	41.04 (33.03–45.99)	58.10 (53.69–62.04)
	(34.51–56.60)	(33.84–58.72)	(54.55–54.67)		
**Hormone secretion (valid cases: 105)**					
No secretion, *n* (%)	76 (72.4)	32 (42.1)	44 (57.9)	29 (65.9)	15 (34.1)
Cortisol/Cortisol + other steroids, *n* (%)	24 (22.9)	6 (25)	18 (75)	11 (61.1)	7 (38.9)
Other secretion, *n* (%)	5 (4.76)	0	5 (100)	4 (80)	1 (10)
**Margins, R status**					
R0, *n* (%)	93 (83)	33 (35.5)	60 (64.5)	40 (66.7)	20 (33.3)
R1, *n* (%)	19 (17)	6 (31.6)	13 (68.4)	9 (69.2)	4 (30.8)
RX, *n* (%)	0	0	0	0	0
**Recurrence-free survival (months)**					
Median (IQR)	26 (12–67.75)	44 (14.75–70.25)	21.5 (11–59.25)	22 (11.5–56)	21 (10.5–71)
**Overall survival (months)**					
Median (IQR)	57.5 (33–88.75)	67.5 (34.75–97)	52.5 (32.5–87.25)	50 (33–110.5)	58 (30.5–82)

**Table 2 biomedicines-10-01873-t002:** Lipid levels and pharmacokinetic data of the 551 samples.

Variable	All (*n* = 551)	Males	Females (*n* = 336)	Premenopausal Women (*n* = 246)	Postmenopausal Women (*n* = 90)
		(*n* = 215)			
**Total cholesterol (mg/dL)**					
Median (IQR)	249.0	247.0	251.0	247.50 (219.0–288.0)	264.50 (234.0–311.50)
	(218.0–283.0)	(209.50–272.50)	(222.0–292.0)		
**HDL (mg/dL)**					
Median (IQR)	71.0	54.0	85.50	88.0 (68.0–107.0)	81.0 (57.0–94.50)
	(53.0–95.0)	(45.0–69.0)	(64.50–103.0)		
**LDL (mg/dL)**					
Median (IQR)	142.80	114.60	140.70	134.60 (102.60–183.60)	147.60 (125.5–197.30)
	(113.0–177.60)	(116.60–170.20)	(110.10–186.70)		
**Triglycerides (mg/dL)**					
Median (IQR)	125.0	157.0	114.50	107.0 (81.0–139.0)	130.50 (97.50–192.50)
	(93.0–179.0)	(110.0–224.0)	(84.0–150.50)		
**Mitotane concentration (µg/mL)**					
Median (IQR)	11.58	11.93	11.22	12.08 (7.25–15.05)	9.88 (4.15–14.11)
	(7.04–15.35)	(7.45–16.35)	(6.41–14.92)		
**o,p’-DDE concentration (µg/mL)**					
Median (IQR)	1.29	1.33	1.18 (0.31–2.35)	1.33 (0.47–2.43)	0.82 (0.09–1.25)
	(0.37–2.44)	(0.47–2.43)			

List of abbreviations: *n*, number; IQR, interquartile range; %, percentage; HDL, high-density lipoprotein; LDL, low-density lipoprotein.

**Table 3 biomedicines-10-01873-t003:** Statistically significant results obtained with Pearson correlation test.

	Mitotane Levels	o,p’-DDE Levels
**Males**	Total cholesterol r = 0.192; *p* = 0.005; IC95%: 0.059–0.317 HDL r = 0.337; *p* < 0.001; IC95%: 0.213–0.450 LDL r = 0.281; *p* < 0.001; IC95%: 0.152–0.399	HDL r = 0.405; *p* < 0.001; IC95%: 0.287–0.511 LDL r = −0.180; *p* = 0.008; IC95%: −0.306–−0.047
**Females**	Total cholesterol r = 0.114; *p* = 0.036; IC95%: 0.007–0.219 HDL r = 0.470; *p* < 0.001; IC95%: 0.382–0.549 Triglycerides r = 0.107; *p* = 0.005; IC95%: 0–0.212	Total cholesterol r = −0.195; *p* < 0.001; IC95%: −0.296–−0.09 HDL r = 0.524; *p* < 0.001; IC95%: 0.441–0.597 LDL r = −0.366; *p* < 0.001; IC95%: −0.455–−0.269 Triglycerides r = −0.206; *p* < 0.001; IC95%: −0.306–−0.101
**Premenopausal woman**	HDL r = 0.495; *p* < 0.001; IC95%: 0.394–0.584	Total cholesterol r = −0.154; *p* = 0.016; IC95%: −0.274–−0.029 HDL r = 0.505; *p* < 0.001; IC95%: 0.406–0.593 LDL r = −0.344; *p* < 0.001; IC95%: −0.450–−0.229 Triglycerides r = −0.180; *p* = 0.005; IC95%: −0.298–−0.056
**Postmenopausal women**	HDL r = 0.359; *p* < 0.001; IC95%: 0.161–0.528 Triglycerides r = 0.328; *p* = 0.002; IC95%: 0.127–0.503	Total cholesterol r = −0.272; *p* = 0.010; IC95%: −0.455–−0.066 HDL r = 0.553; *p* < 0.001; IC95%: 0.389–0.683 LDL r = −0.434; *p* < 0.001; IC95%: −0.590–−0.247

List of abbreviations: IC, interval of confidence at 95%; HDL, high-density lipoprotein; LDL, low-density lipoprotein.

## Data Availability

Not applicable.

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
