# Peer review of "Sex-Based Evaluation of Lipid Profile in Postoperative Adjuvant Mitotane Treatment for Adrenocortical Carcinoma"

_biomedicines, 2022, doi:10.3390/biomedicines10081873_

Round 1

Reviewer 1 Report

Dear Authors,

I read the manuscript entitled "Gender-based evaluation of the effect of mitotane on total cholesterol, HDL, LDL and triglycerides levels in patients with adrenocortical carcinoma." The paper focuses on an interesting subdomain of mitotane that is indicated for the symptomatic treatment of adrenal cortical carcinoma (CCS). Numerous papers have been written in the literature on the undesirable effects of mitotane on the lipid profile. However, this paper is the first to raise the issue of gender difference in the case of mitotane-induced dyslipidemias.

The authors of the study have a solid knowledge of the pharmacokinetics and pharmacodynamics of drugs, gender being an important factor that could influence the plasma concentration of mitotane with the risk of dyslipidemia. Furthermore, we noted that the authors' concern about gender was also addressed in a paper "Sex Differences on Mitotane Concentration and Treatment Outcome in Patients with Adrenocortical Carcinoma", which shows that between females and males, there are differences in the kinetics of a drug.

The paper is presented in a clear and concise manner, respecting the requirements of the journal. I appreciate the fact that the authors also gave us some conclusions.

The bibliographic references are in acceptable numbers, some of them being as recent studies as possible.

The results were presented in the form of graphs and a table, being easy to read.

I have only a few minor comments:

  1. Among the female population, you compared the premenopausal (246) and postmenopausal (90) groups. Could the authors argue whether the different number of patients in the groups could have influenced the results of the study?
  2. In the introductory part you mentioned that mitotane acts through an adrenolytic mechanism. Could you present some information about this mechanism in the Discussions section?
  3. The English language, although quite good, could be improved.

Author Response

Thank you for your interesting comments about our paper. Following the effected changes and the discussed points:

1. 

Thank you for the professional observation. As you can see the numerousness of the compared groups is different. For this reason, we have chosen to use the non-parametric Mann-Whitney test to compared them; this test compares the mean ranks and not medians and distributions. The obtained P value shows the chance that a randomly selected value from the population with the larger mean rank is greater than a randomly selected value from the other population. We added this information in the Materials and Methods section.

2. Thank you for your comment. Information about the adrenolytic mechanism has been inserted in the Discussion section, as you suggested.

3. Thank you, the text has been carefully revised according to your suggestion.

Reviewer 2 Report

The presentation of the results in the article is illegible. The levels of LDL, HDL, and holesterol in the charts are a repetition of the results from the table.

The correlation between mitotane and blood lipid levels should be presented in the tables.

Only patients with complete tumor resection were considered in the studies. This should be more emphasized in the text or in the title

Blood lipids should be related to healthy people

The name of the study drug should be standardized throughout the article. In the title we have mitotane, in the results o, p'-DDD, both names in the discussion.

The drug was administered "up to 8-10 g daily or the maximum tolerated dose". Dosing could be related to a patient parameter that is also related to lipid levels. Authors should show the relationship between a drug dose and blood lipid levels and changes in lipid levels (start of treatment, end of treatment)

Author Response

Thank you for your interesting comments about our paper. Following the effected changes and the discussed points:

  1. Thank you for your comments. The Results section has been revised.
  2. Thank you for your suggestion, the Table (Table 2) has been inserted and the Results section revised.
  3. Thank you for your suggestion, this information has been emphasized in the text, has you recommend.
  4. Thank you for your professional comment. We asked our Local Ethics Committee for the approval of a new protocol that also provides for the enrollment of healthy volunteers with whom to compare the levels of blood lipids. This topic will be the subject of our future work. However, we added this lack as study limitation.
  5. Thank you for your comments. The name of the drug (mitotane) has been standardized throughout the article.
  6. Thank you for the interesting comment. This study protocol did not provide for the recording of the data before the start of the therapy, so we only used the data three months after the start of the therapy, before dose changes were made and / or the statin was taken. We added this information in Materials and Methods section.

Reviewer 3 Report

The study is interesting by assessing lipid balance in relation to sex and pre- or postmenopausal status
Some comments
-specify what type of hormones the carcinoma produced and whether both adrenal and gonadal hormone status was assessed after surgery. Many excess hormones can affect cholesterol and triglyceride secretion (e.g., estrogen, cortisol, testosterone, or other androgens in both male and female)
-Specify whether in non-menopausal women there was previously amenorrhea and whether the cycle resumed after surgery and during mitotane
-Specify whether any patients had diabetes or hypertension and what therapy was administered before and during the ongoing mitotane evaluation
-Discuss a possible role of the effect of mitotane on SSHBG and CBG on the òlipid measurement
-specify whether resection was complete as this is rare in adrenal carcinoma. Report whether there was tumor recurrence remotely after surgery.
-Report whether thyroid hormones were measured before surgery and during mItotane. The drug may create hypothyroidism

Author Response

Reviewer #3:

The study is interesting by assessing lipid balance in relation to sex and pre- or postmenopausal status

Some comments

Thank you for your interesting comments about our paper. Following the effected changes and the discussed points:

-specify what type of hormones the carcinoma produced and whether both adrenal and gonadal hormone status was assessed after surgery. Many excess hormones can affect cholesterol and triglyceride secretion (e.g., estrogen, cortisol, testosterone, or other androgens in both male and female)

Thank you for your professional comment. The patients included in the study were all in adjuvant treatment with mitotane, thus free of disease. For this reason, our results were not influenced by tumour hormonal secretion, as explained in discussion section. However, we added information about tumour secretion at diagnosis in table 1.

-Specify whether in non-menopausal women there was previously amenorrhea and whether the cycle resumed after surgery and during mitotane

Thank you for your interest. Unfortunately, we collected only the information on FSH and LH levels to determine the reach of menopausal period. We clarify this in materials and methods section. However, as reported in another paper of our group (Basile et al. doi 10.3390/cancers12092615), menstrual cycles were usually maintained in our female patients during mitotane therapy.

-Specify whether any patients had diabetes or hypertension and what therapy was administered before and during the ongoing mitotane evaluation

Thank you for your comment. Patients with history of other previous/concomitant diseases and use of other medications interfering with mitotane effect were excluded from the study. We clarify this in materials and methods section.

-Discuss a possible role of the effect of mitotane on SSHBG and CBG on the lipid measurement

Thank you for your suggestion. This study protocol does not include the measurement CBG levels, but only of SHBG levels, as added in materials and methods section. Moreover, we inserted the effect of mitotane on CBG and SHBG in the introduction section.

-specify whether resection was complete as this is rare in adrenal carcinoma. Report whether there was tumor recurrence remotely after surgery.

Thank you for your comment. All included patients were treated with adjuvant mitotane, thus at the time of drug dosing, none of them had tumour recurrence. We specify this in materials and methods section. In addition, we inserted in the table 1 also recurrence free survival and overall survival information and R status.

-Report whether thyroid hormones were measured before surgery and during mItotane. The drug may create hypothyroidism

Thank you for your interest. As per clinical practice, thyroid hormones were periodically measured and patients with hypothyroidism were adequately replaced with LT4 therapy, therefore avoiding influence on lipid profile. We added this information in materials and methods section.

Reviewer 4 Report

Allegra et al. present an interesting study on the gender-specific effect of mitotane on the lipoprotein and lipid levels in patients with adrenocortical carcinoma. However, I am not sure if the results can be interpreted as described in the manuscript. In my opinion, the design of the study is flawed.

 The authors do not mention when (in which phase of the treatment) the lipoprotein and lipid levels were measured. Moreover, since the lipoprotein and lipid levels were not measured before, during, and after the mitotane treatment, it cannot be concluded that their levels are associated with the treatment. Sure, there are gender-based differences in the lipoprotein and lipid levels - but this is the only conclusion that can be drawn from the presented results.

Author Response

Reviewer #4:

Allegra et al. present an interesting study on the gender-specific effect of mitotane on the lipoprotein and lipid levels in patients with adrenocortical carcinoma. However, I am not sure if the results can be interpreted as described in the manuscript. In my opinion, the design of the study is flawed.

Thank you for your interesting comments about our paper. Following the effected changes and the discussed points:

The authors do not mention when (in which phase of the treatment) the lipoprotein and lipid levels were measured.

Moreover, since the lipoprotein and lipid levels were not measured before, during, and after the mitotane treatment, it cannot be concluded that their levels are associated with the treatment.

Sure, there are gender-based differences in the lipoprotein and lipid levels - but this is the only conclusion that can be drawn from the presented results.

Thank you for your professional suggestion. In this study we measured only the levels of total cholesterol, HDL, LDL and triglycerides at the time of mitotane dosing, to underline sex-based differences in our cohort of patients treated with mitotane. Further studies, including the lipid measurement before and after mitotane starting treatment will be performed by our group. The title, the aim and the conclusions of the manuscript have been consequently modified.

Round 2

Reviewer 2 Report

The authors did not adapt the article to the reviewer's guidelines. All data must be checked against the control (healthy people).

The results also need to be compared with the patients before treatment. They can be completely different patients. Showing only that the patients after the therapy have a given cholesterol level is incorrect. Different populations will have different cholesterol levels due to, for example, diet. Someone else from another country with an obesity epidemic will compile data from a peer-reviewed article and conclude that mitotane patients have lower cholesterol levels than healthy people.

Author Response

Dear Reviewer, thank you for your professional comment. The comparisons of the lipid levels between treated patients and healthy volunteers or between before and after the start of therapy are not included in this study protocol, already approved from the Local Ethic Committee. We added these lacks in the study limitations. Moreover, to meet your request, we have included the reference value of HDL, LDL, triglycerides and total cholesterol, in the materials and methods.

Eventually, we would like to underline that the objective of our study is not to evaluate the increase in blood lipid levels, but rather to describe the levels of these in the different populations under study: males, females, premenopausal women and postmenopausal women. A further goal was to describe the correlations (Pearson's test) between blood lipids and drug levels, not to define a causal relationship between the variables. Instead our conclusion is: The obtained results suggest that a gender and personalized approach should be useful to prevent and to better control dyslipidemias.

Reviewer 4 Report

The authors of the study implementer suggested changes to their manuscript which led to its significant improvement. Now the paper is clearer and concise and the limitations of the study are also explained.  The manuscript, however, still needs some English language editing.

Round 3

Reviewer 2 Report

The authors did not adapt the article to the reviewer's guidelines. All data must be checked against the control (healthy people).

The results also need to be compared with the patients before treatment. They can be completely different patients. Showing only that the patients after the therapy have a given cholesterol level is incorrect. Different populations will have different cholesterol levels due to, for example, diet. Someone else from another country with an obesity epidemic will compile data from a peer-reviewed article and conclude that mitotane patients have lower cholesterol levels than healthy people.

Author Response

Thank you for your professional suggestion. In this study we measured only the levels of total cholesterol, HDL, LDL and triglycerides at the time of mitotane dosing, to underline sex-based differences in our cohort of patients treated with mitotane. Further studies, including the lipid measurement before and after mitotane starting treatment will be performed by our group. The title, the aim and the conclusions of the manuscript have been consequently modified.